# Fabrication and Tribology Properties of Cr-Coated Cemented Carbide under Dry Friction Conditions

**Li Zhang, Wenlong Song \*, Lei An, Zixiang Xia, Shoujun Wang and Tianya Li**

School of Industry, Jining University, Qufu 273155, China; qufuzhangli@jnxy.edu.cn (L.Z.);
anlei@jnxy.edu.cn (L.A.); xiazixiang168@163.com (Z.X.); shoujun0531@163.com (S.W.); lt1990@jnxy.edu.cn (T.L.)
\* Correspondence: wlsong@jnxy.edu.cn

**Abstract:** To improve the surface friction characteristics of cemented carbide, a Cr coating was deposited on cemented carbide substrate with the multiple arc plating technique. The surface and cross-section micrographs, adhesion force and micro-hardness of the Cr-coated cemented carbide were tested. The anti-friction and anti-wear behaviors of cemented carbide with and without Cr coating were investigated via the sliding friction test against a WC/Co ball. The tribological properties of cemented carbide were examined using a scanning electron microscope and energy dispersive X-ray analysis. The test results showed that Cr-coated cemented carbide possessed good adhesion properties and lower hardness. The average coefficient of friction for Cr-coated cemented carbide was reduced by 10–20% compared with that of an uncoated one. The primary wear modes of the Cr-coated sample were delamination of coating, flaking and abrasion wear. It can be found that the preparation of a Cr coating is an effective way to enhance the friction and wear performance of traditional cemented carbide.

**Keywords:** tribological properties; Cr coatings; cemented carbide; multi-arc ion plating

## 1. Introduction

Due to the characteristics of high toughness, excellent thermal stability, and superior tribological performance [1], cemented carbide has been widely utilized in cutting tools, engine parts, sealing element and bearing components [1,2]. However, without the cooling and lubrication of a cutting fluid, the carbide exhibits relatively high friction, serious adhesion and a short lifetime.

Surface coating technology is a promising way to reduce the friction and wear of substrate material under different working conditions. Surface coating can significantly enhance the load-bearing capacity and area of application of a substrate surface. Based on the surface micro-hardness, surface coating can be divided into hard coatings and soft coatings. Hard coatings comprise wear-resistant coatings, which are mainly composed of one- and multi-component transition metal nitrides or oxides. Hard coatings have a high surface hardness and strength, good abrasion resistance, and high thermal stability [3–5], and have been widely applied in tough work environments, such as dry machining, high-speed turning and stamping molds [6–10]. However, hard coatings are still subject to high mechanical stress and high temperature adhesion wear in rough working conditions. In addition, they also display a high friction coefficient during the dry friction process, which could lead to the reduction in wear resistance and service life.

Soft coatings are efficient in improving the surface lubricity and reducing the friction via forming a lubricating film on the contact area [11]. The properties of soft coatings on cemented carbide have been studied deeply by experts and scholars. Tungsten disulfide ($WS_2$) [12–15] and molybdenum disulfide ($MoS_2$) [16–20] are the most widely used soft coatings in industrial applications. These sulfides have a very low friction coefficient owing to their lamellar structure, where a layer of metal atom is sandwiched between double

hexagonal sulfur layers. However, sulfide soft coatings are limited due to their characteristics of environmental sensitivity and low hardness. To improve the wear characteristics and working time of sulfide coatings, the combination of sulfides and metals (e.g., Ti, Cr, C, Ag, Ni or Zr) has been reported to evidently improve the mechanical and tribological performance [21–31], and has been largely used in various industrial applications.

Nevertheless, the current research mainly focuses on hard coatings and soft coatings containing metal combinations (for instance, Cr, Ti, and Zr). Research on the characteristics of single metal-coated cemented carbide is still highly lacking [32–35]. It is necessary to comprehensively study the tribological behavior of pure metal-coated carbide to expand the application range.

To further improve the interfacial adhesive properties between the substrate surface and coating, pre- and post-treatments technologies, such as sandblasting treatment, electrochemical corrosion, nitriding treatment, and laser treatment, are employed to activate and purify the material substrate surface [36–39]. And among them, laser treatment has competitive advantages in coatings preparation, being able to produce geometric textures with different morphologies, increase the surface contact area, and provide a good adhesion interface and mechanical locking ability for the coatings.

In this research, a Cr coating was prepared on cemented carbide (WC + 14%TiC + 6%Co) by means of the multiple arc plating method. The mechanical and physical properties of Cr coating were evaluated, and the friction behavior of Cr-coated carbide against WC ball was also investigated. This work can improve the industrial application of cemented carbide.

## 2. Materials and Methods

### 2.1. Cr Coating Preparation

Cemented carbide was used as the sample material. The main mechanical and physical properties of cemented carbide are listed in Table 1. The carbide sample surface was burnished to a mirror finish. The Cr coating was prepared by means of the multiple arc ion plating technique utilizing two pure Cr targets (99.999%). The depositing parameters of the Cr coating are indicated in Table 2. Figure 1 indicates the optical photos of the traditional sample and Cr-coated one.

**Table 1.** Mechanical properties of cemented carbide.

| Composition (wt. %) | Density (g/cm$^3$) | Hardness (GPa) | Flexural Strength (MPa) | Young's Modulus (GPa) | Thermal Expansion Coefficient ($10^{-6}$/K) | Poisson's Ratio |
|---|---|---|---|---|---|---|
| WC + 14%TiC + 6%Co | 11.6 | 15.4 | 1200 | 510 | 6.50 | 0.25 |

**Table 2.** Deposition parameters of the Cr coating.

| Substrate | Base Pressure (Pa) | Temperature (°C) | Ar Pressure (Pa) | Cr Current (A) | Depositing Temperature (°C) | Deposition Time (min) |
|---|---|---|---|---|---|---|
| Cemented carbide | $6.0 \times 10^{-3}$ | 250 | 0.6 | 85 | 220 | 150 |

The scratch test has been proven to be effective in evaluating the adhesion between a surface coating and substrate. An MFT-4000 scratch tester (Lanzhou Instrument Technology Co., Ltd., Lanzhou, China) was employed to measure the adhesion force of the Cr coating with carbide via scratching a diamond indenter with a radius of 200 μm along the Cr coating's surface with a gradually increasing force. The test conditions for adhesion force are shown below: a scratching distance of 8 mm, a loading force range of 0–100 N. During the scratch test, the friction force and acoustic signal were collected automatically. The slope

pivotal points were obtained from the curves of friction force and acoustical signal versus scratch load, and were determined as the adhesion force corresponding to the continuous coating cracking and substrate exposure.

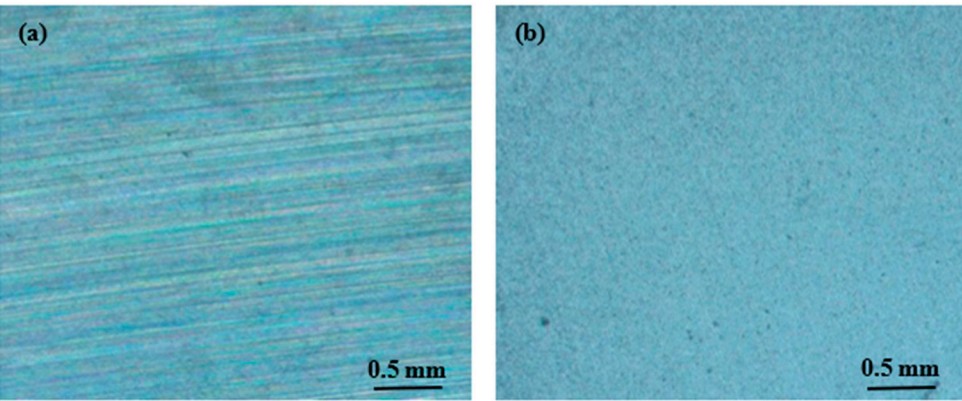

**Figure 1.** Optical micrographs of the uncoated sample (**a**) and Cr-coated sample (**b**).

The surface micro-hardness of the Cr coating was tested on a MH-6 micro-hardness meter (Shanghai Testing Instrument Co., Ltd., Shanghai, China) with a loading force of 0.2 N.

*2.2. Friction Tests*

The anti-friction and anti-wear properties were tested by means of a HRT ball-on-disk tribometer (Jinan Hengxu Testing Machine Technology Co., Ltd., Jinan, China). The upper sample was a WC/Co ball (WC + 6%Co) with a diameter of 9.5 mm, and the sample below was Cr-coated cemented carbide with dimensions of 16 mm × 16 mm × 5 mm. The Cr-coated carbide was fixed, and the WC ball above was slid reciprocally against the lower carbide. The sliding friction tests were carried out with the parameters below: loading force $F$ = 20–80 N, sliding speed $v$ = 4–10 mm/s, stroke length $l$ = 6 mm. The duration of all of the sliding tests was 900 s. The friction tests were carried out at a room temperature of 22–25 °C, and an air humidity of 35–48%. The friction coefficient was collected automatically using the tribometer. All of the results were the average of three tests in order to guarantee the stability and repeatability of test data. After sliding tests, the worn coated carbide was examined using a scanning electron microscope (SEM, INCA Penta FETXS, Oxford, UK), and the chemical composition at the worn area was tested via energy dispersive X-ray analysis (EDX, D8 ADVANCE, Bruker, Germany)

**3. Results and Discussion**

*3.1. Properties of the Cr-Coating*

Figure 2 shows the surface micrograph and cross-section micrograph of Cr-coated carbide. As shown in Figure 2a, it was obvious that the Cr coating was relatively uniform and dense. However, the Cr coating's surface was a bit rough, and a lot of micro-particles could be observed on the surface. This is very common in coatings fabricated by means of multiple arc ion plating technique. As seen in Figure 2b, the Cr coating closely stuck to the carbide substrate, there were no significant cracks or delamination defects at the coating–substrate interface, and the thickness of the Cr coating was about 7.5 ± 0.2 μm.

The adhesion force between the Cr coating and cemented carbide substrate can be obtained based on the curve variations of friction force and acoustic signal. Figure 3 indicates the scratch curve of the Cr-coated carbide. In the initial stage, the value of the acoustic signal was small enough to be negligible, and the rubbing force curve was relatively sleek and smooth. As scratch force gradually grew, the fluctuations of friction force and sound signal increased greatly. When the force was above 75 N, the variation of the signal curve reached the maximum, which reflected that the Cr coating was entirely worn out.

Therefore, the adhesion force between the Cr coating and substrate was determined to be about 75 N.

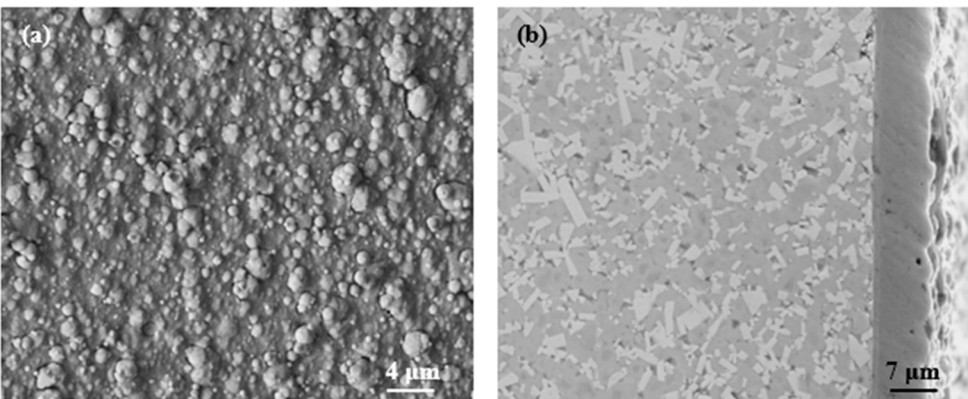

**Figure 2.** Surface micrograph (**a**) and cross-section micrograph (**b**) of the Cr-coated carbide.

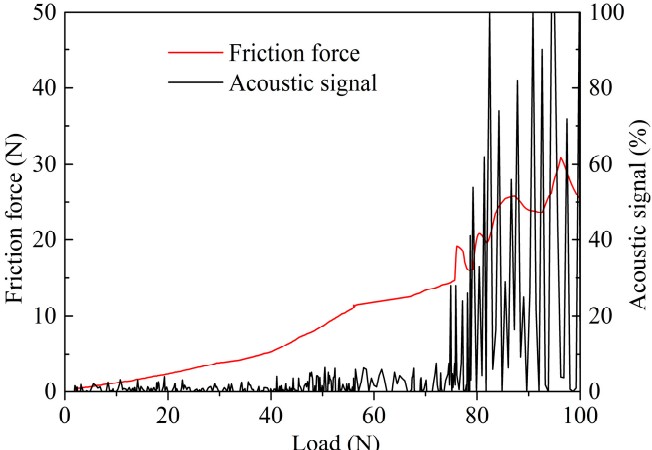

**Figure 3.** Scratch curve of adhesion strength test for the Cr-coated sample.

In order to better evaluate the adhesion force, the SEM micrographs and EDX results of the scratched surface are indicated in Figure 4. As indicated in Figure 4, it can be found that the scratch test of the Cr coating fall into two stages. In the initial stage, there were obvious scratch ploughs, and the Cr coating was still on the scratch surface, which was confirmed by the micrograph and EDX of point A in Figure 4b,d. This led to the lower friction force and sound signal (see Figure 3). With the increase in loading force, the Cr coating was gradually worn down, and the carbide substrate was exposed, which was confirmed by the micrograph and EDX of point B (Figure 4c,e). Therefore, the curve fluctuations of the test signal were improved rapidly. The micrographs and EDX results were in agreement with the change in the scratch test curve in Figure 3.

The micro-hardness, thickness and adhesion strength of Cr-coated carbide are listed in Table 3. This indicates that the micro-hardness of Cr-coated carbide was just $13.2 \pm 0.5$ GPa, which was reduced by about 15.5% compared with that of uncoated carbide ($15.7 \pm 0.5$ GPa).

### 3.2. Tribological Performance of Cr-Coated Carbide

Figures 5 and 6 show the average value of the friction coefficient for the Cr-coated sample and the uncoated one under different sliding conditions. As can be seen, the friction coefficient of the Cr-coated sample was about 10–20% lower than that of the uncoated one at the same sliding conditions. In Figure 5, when the speed of sliding gradually increased, the average value of the friction coefficient for the Cr-coated sample was about 0.21–0.23,

while the friction coefficient of the uncoated sample was about 0.25–0.26. As indicated in Figure 6, it can be found that the value of the friction coefficient slightly decreased with the increasing applied force. When the loading force increased from 20 N to 80 N, the average friction coefficient of the coated carbide reduced from 0.236 to 0.217, while that of the uncoated carbide decreased from 0.263 to 0.251. The results of the sliding test above demonstrate that a Cr coating can contribute to a decrease in the friction coefficient for cemented carbide.

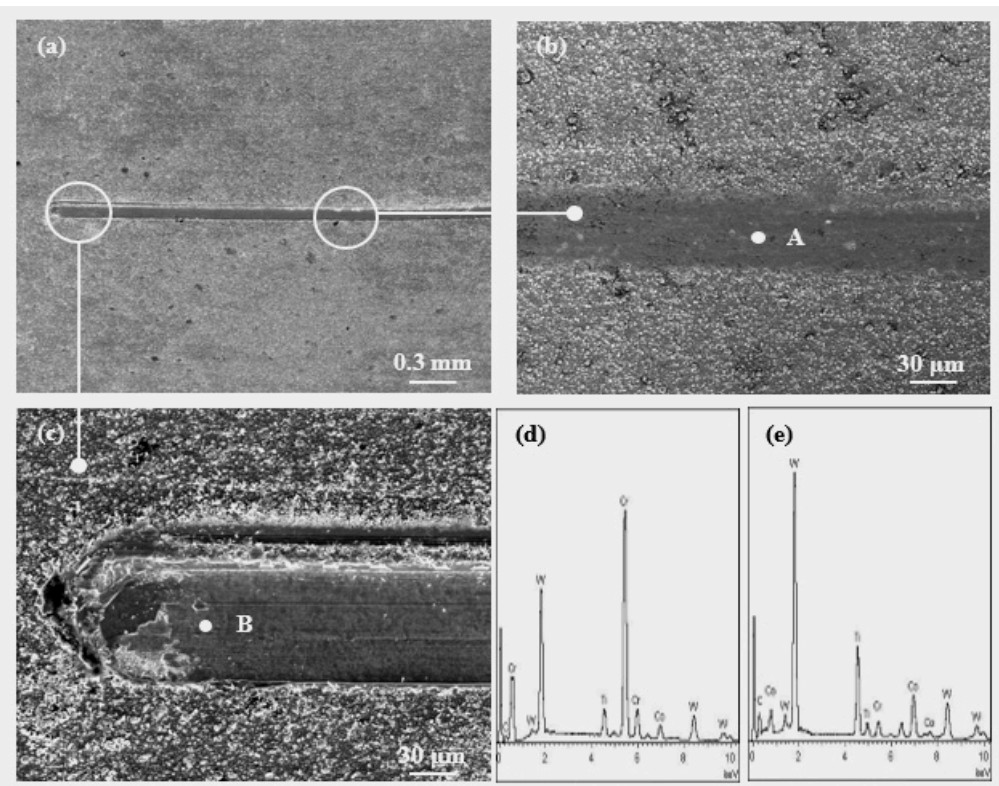

**Figure 4.** Scratch micrographs (**a–c**) and corresponding EDX results of points A (**d**) and B (**e**).

**Table 3.** Mechanical properties of the Cr coating.

| Substrate | Coating | Micro-Hardness (GPa) | Thickness (μm) | Adhesion Strength (N) |
|---|---|---|---|---|
| Cemented carbide | Cr | 13.2 ± 0.5 | 7.5 ± 0.2 | 75 ± 5 |

The average value of the friction coefficient for frictional pairs under elasticity load conditions can be expressed by the formula below [32]:

$$\mu = \tan \beta = \frac{F_f}{P} = \frac{\overline{\tau}_c A_r}{\sigma_b A_r} = \frac{\overline{\tau}_c}{\sigma_b} \tag{1}$$

where $\beta$ is the friction angle, $P$ is the applied force, $F_f$ is the friction force, $\sigma_b$ is yield stress of sample, $A_r$ is the real contact area, and $\overline{\tau}_c$ is the average shear stress of the sample.

In general, the yield ultimate stress of the cemented carbide substrate remains almost the same [32]. According to the above formula, the reduced average shearing stress of the carbide surface is beneficial to decrease surface friction. Because the shearing stress of the Cr coating is obviously smaller than that of cemented carbide, Cr-coated cemented carbide is helpful to reduce the average friction coefficient, and this is consist with the test curve of the friction coefficient indicated in Figures 6 and 7.

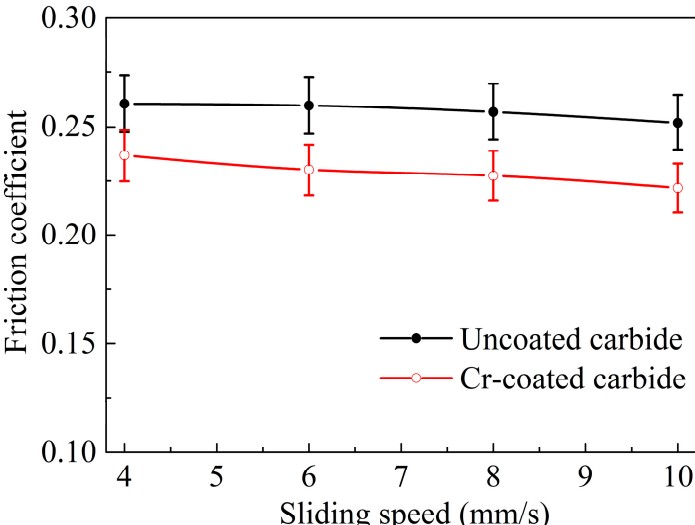

**Figure 5.** Average value of the friction coefficient at different sliding speeds (load = 60 N, sliding time 15 min).

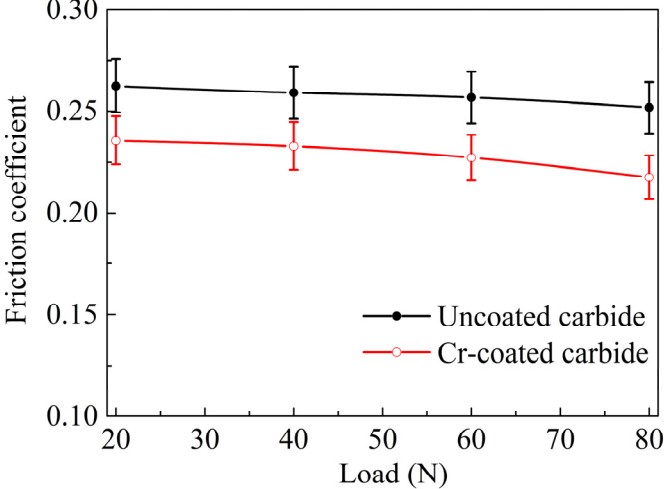

**Figure 6.** Average value of the friction coefficient at different loads (sliding speed = 8 mm/s, time 15 min).

*3.3. Tribological Morphologies*

The tribological properties of the tested sample in the sliding process can be determined via SEM and EDX. Figure 7 shows the SEM micrographs and EDX results in the worn surface of uncoated carbide (loading force = 60 N, speed = 8 mm/s, friction time = 15 min). As indicated in Figure 7a,b, abrasion wear and adhesion wear were clearly found on the wear track. The EDX results (Figure 7c,d) proved that there were WC elements of a sliding pair. The continuous friction and squeezing between the friction pairs deteriorated the surface finish and aggravated the abrasion wear of the uncoated surface, which led to serious micro-cracks and mechanical scratches on the worn track of uncoated carbide (see Figure 7b).

The SEM micrographs and EDX results of the Cr-coated carbide are shown in Figure 8. As indicated in Figure 8a,b, serious coating flaking and delamination were found on the worn surface due to the brittle fatigue fracture and the continuous sliding friction. The corresponding EDX results on the worn area are indicated in Figure 8c,d. It was easy to see that there also existed some adhesion wear, and there were no any obvious micro-cracks or mechanical scratches on the surface of the sample because of the protection of the Cr coating. Therefore, the primary wear modes of the Cr-coated carbide were coating flaking,

delamination and abrasion wear, and the Cr coating was helpful to decrease the abrasion wear and adhesion wear of cemented carbide. It also can be found that the tribological properties rely on the material property of friction pairs and the experimental conditions.

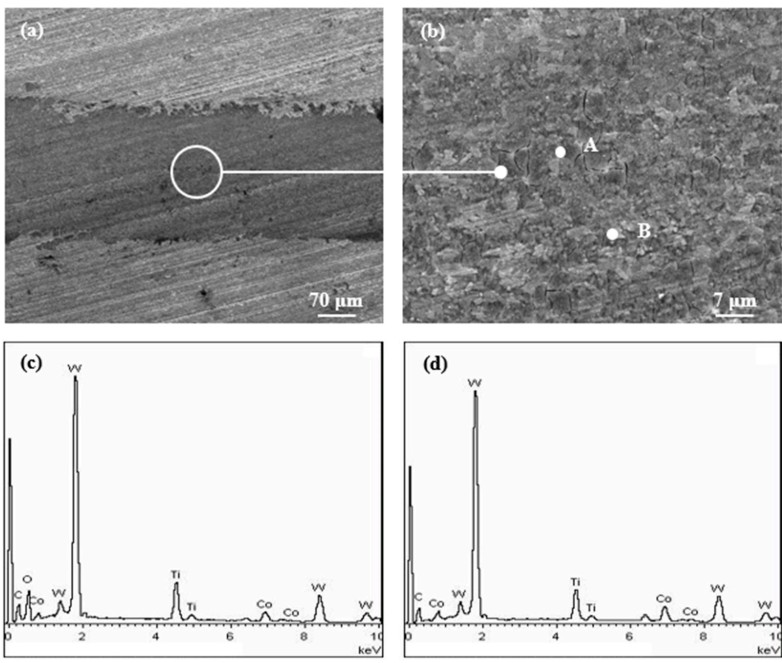

**Figure 7.** SEM micrographs and EDX results in worn surface of uncoated carbide: (**a**) worn surface, (**b**) enlarged micrograph and EDX results of point A (**c**) and B (**d**) in (**b**) (load = 60 N, speed = 8 mm/s, time 15 min).

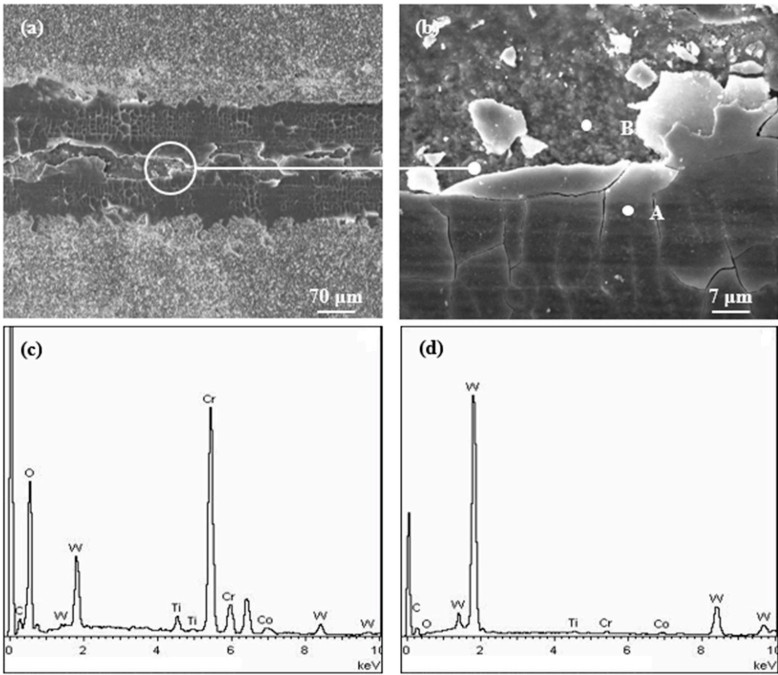

**Figure 8.** SEM micrographs and EDX results for the worn surface of the Cr-coated carbide: (**a**) worn surface, (**b**) enlarged micrograph and EDX results of point A (**c**) and B (**d**) in (**b**) (load = 60 N, speed = 8 mm/s, time 15 min).

Future investigation work will be carried out on the mechanical and tribological properties of Cr coatings under different pretreatments, post treatments, friction models and other severe conditions (temperature, humidity, etc.).

## 4. Conclusions

A pure Cr coating was fabricated on a cemented carbide surface using the multiple arc ion plating technique. The mechanical properties and friction performance of the Cr-coated sample were compared with those of an uncoated samples. The primary conclusions can be obtained as follows:

1.  The pure Cr coating fabricated on the carbide surface exhibited good adhesion force. The adhesion force of the Cr coating was about 75 N. The micro-hardness of the Cr-coated carbide was about 13.2 GPa, and the thickness was about 7.5 μm.
2.  The average value of the friction coefficient for Cr-coated carbide was reduced by about 10–20% compared with that of the uncoated one under the exact same conditions. The average friction coefficient reduced with the increase in loading force, and it varied lightly with different friction speed.
3.  The surface Cr coating can reduce the wear and tear of traditional cemented carbide, and the primary wear modes of the Cr coating were flaking of the coating, delamination and abrasion wear. A Cr coating can be considered as an effective way to enhance the friction and wear performance of traditional cemented carbide.

**Author Contributions:** W.S. and Z.X. conceived and designed the experiment; W.S. and S.W. performed the experiment; L.A., Z.X. and T.L. analyzed the data; W.S. and L.Z. wrote the paper. All authors have read and agreed to the published version of the manuscript.

**Funding:** This research was funded by the Key Research and Development Program of Jining of China (2022HHCG014), the Scientific Research Foundation of Jining University (2021CGZH01, 2022HHKJ02) and the China Postdoctoral Science Foundation (2021M701408).

**Data Availability Statement:** Not applicable.

**Conflicts of Interest:** The authors declare no conflict of interest.

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
