# Peer review of "Fabrication and Tribology Properties of Cr-Coated Cemented Carbide under Dry Friction Conditions"

_lubricants, doi:10.3390/lubricants11070278_

Round 1

Reviewer 1 Report

Please, see the file attached with my comments.

Author Response

Dear Reviewer:
Thank you for your letter and for the reviewer’ comments concerning our manuscript entitled “Fabrication and tribology property of Cr-coated cemented carbide under dry friction conditions” (ID: lubricants-2367373). These comments are all valuable and very helpful for revising and improving our paper, as well as the important guiding significance to our researches. We have studied comments carefully and have made corrections which we hope meet with approval. In order to clearly show the changes of the manuscript to the editors and reviewers, the function of "Track Changes" in Microsoft Word was used, and the revised manuscript can be displayed properly by clicking the final status without markup in the review tab. The main corrections in the paper and the responds to the reviewer’s comments are as flowing:

Comments and Suggestions for Authors

A multiple arc ion plating technique was employed to create a pure Cr coating on a cemented carbide surface. The mechanical properties and friction performance of the Cr-coated sample were compared with those of the uncoated one. The main findings are as follows:

The pure Cr coating on the carbide surface demonstrated good adhesion force, with a strength of approximately 75 N. The micro-hardness of the Cr-coated carbide was around 13.2 GPa, and the coating thickness was about 7.5 μm. However, the pretreatment and post treatments were not well documented and discussed, as it was in the better source for this, see htps://doi.org/10.1080/10426914.2014.973582 because pre and post-treatments are key in adhesion of the new layers of the final performance of them.

Under identical conditions, the average friction coefficient for the Cr-coated carbide was 10% to 20% lower than that of the uncoated one. The average friction coefficient decreased with increasing loading force and varied slightly with different friction speeds. The same happened in some severe cases of friction in cutting, see Fox work https://doi.org/10.1016/S0257-8972(99)00611-8 and others that used the same ideas. htps://doi.org/10.1016/j.jmapro.2020.08.029.

The ideas could be better discussed, use above information: pre and post-treatment, severe conditions. Etc. Cu is soft and perhaps you are thinking in self-healing coatings. The surface Cr coating reduced the wear and tear of traditional cemented carbide, with the primary wear being significantly reduced.

Response:

This manuscript just mainly aims at the fabrication and tribological property of Cr-coated cemented carbide. The surface and cross-section micrographs, adhesion force and micro-hardness of the Cr-coated carbide were tested. The friction and wear behaviors of carbide with and without Cr coating were investigated by sliding friction test. The test results showed that Cr-coated carbide possessed good adhesion property and lower hardness. The average friction coefficient of Cr-coated carbide was reduced compared with that of uncoated one. The primary wear modes of Cr-coated sample were coating delamination, flaking and abrasion wear. It can be found that preparation of Cr coating is an effective way to enhance the friction and wear performance of traditional cemented carbide.

Considering the Reviewer’s suggestion, the pretreatment, post treatments and severe conditions are also important factors which affect the property of Cr-coated carbide. Then it is necessary to comprehensively study on the effects of different conditions on the coating property for further research. The statement about the further work was added in the end of section 3.3 as follows:

The future investigation work will be carried out on the mechanical and tribological properties of Cr coating under different pretreatments, post treatments, friction models, heating-friction process and other severe conditions (temperature, humidity, etc.).

Did you use other friction coefficient different to coulomb ones?. Since N. Zorev, many other models are more useful than the simple Amontons-Coulomb one. A better discussion can eliminate the necessity to make a better experimental campaign. See above ideas, because after Zorev the friction models changed: stick and sliding zones are different.

Response:

This manuscript mainly investigated and compared the tribological property of carbide with and without Cr coating at the same sliding conditions based on coulomb model. The test results showed that the average friction coefficient of coated sample was reduced by about 10%-20% compared with that of uncoated one, and the Cr coating can reduce the wear and tear of cemented carbide. The Cr coating can be considered as an effective way to improve the friction and wear performance of traditional cemented carbide. Then the model is not a significant factor about the tribological property of sample in the process of comparing friction performance of sample.

Considering the Reviewer’s suggestion, further research will be conducted on the effects of different friction models on the tribological behavior. The statement about the further work was added in the end of section 3.3 as follows:

The future investigation work will be carried out on the mechanical and tribological properties of Cr coating under different pretreatments, post treatments, friction models, heating-friction process and other severe conditions (temperature, humidity, etc.).

Acoustic signals…why are they interesting? On the other hand, some ideas were defined in the heating-friction process defined in works about Process performance and life cycle assessment of friction drilling on dual-phase steel. Friction dirlling bring coatings to its maximum level. https://doi.org/10.1016/j.jestch.2020.10.001 is other key work, because the journal Engineering Science and Technology, an International Journal is now important in your field.

Response:

The adhesion force between Cr coating and cemented carbide substrate can be determined by the change of friction force and acoustic signal owing to the coating tear and spalling during scratching process. This testing method has been proven effective in measuring adhesion force.

Heating-friction is an important factor which affects the friction property of Cr coating. Then it is necessary to comprehensively study on the effects of different conditions on the coating property for further research.

Considering the Reviewer’s suggestion, further research will be conducted on the effects of heating-friction process on the tribological behavior of Cr coating. The statement about the further work was added in the end of section 3.3 as follows:

The future investigation work will be carried out on the mechanical and tribological properties of Cr coating under different pretreatments, post treatments, friction models, heating-friction process and other severe conditions (temperature, humidity, etc.).

SUMMARY: PAPER NEEDS BETTER DISCUSSION, or as alternative define a better friction model based on regression data including temperature.

Response:

The friction and wear behavior of cemented carbide with and without Cr coating were investigated by sliding friction test against a WC/Co ball at room temperature. The variation of temperature under the sliding conditions is not very obvious (less than 2 ℃), then the manuscript does not contain the test result of temperature.

Considering the Reviewer’s suggestion, further research will be conducted on the effects of different friction models on the tribological behavior. The statement about the further work was added in the end of section 3.3 as follows:

The future investigation work will be carried out on the mechanical and tribological properties of Cr coating under different pretreatments, post treatments, friction models, heating-friction process and other severe conditions (temperature, humidity, etc.).

We tried our best to improve the manuscript and made some changes in the manuscript.  These changes will not influence the content and framework of the paper. And here we did not list the changes but can be tracked down in revised paper by clicking "Track Changes".

We appreciate for your warm work earnestly, and hope that the correction will meet with approval.

Once again, thank you very much for your comments and suggestions.

Best Regards,

Corresponding author: Wenlong Song

Reviewer 2 Report

The paper provides a good overview of the friction behavior of chromium-coated cemented carbide. In this study, a chromium coating was deposited on carbide substrates using a multiple arc ion plating process, and it was examined for adhesion strength, microhardness, and tribological properties.

The paper is well structured and very well written. There are minor spelling errors present, such as in line 86 "there were no obvious," and in line 120, Zr was swapped with Cr. The labels in the graphs should also be highlighted to make them more visible.

The results of the paper are clear and well-developed. However, it would be beneficial to emphasize the ultimate application goal of the coated cemented carbides to make the results more applicable for future use.

There are minor spelling errors present, such as in line 86 "there were no obvious," and in line 120, Zr was swapped with Cr.

Author Response

Dear Reviewer:
Thank you for your letter and for the reviewer’ comments concerning our manuscript entitled “Fabrication and tribology property of Cr-coated cemented carbide under dry friction conditions” (ID: lubricants-2367373). These comments are all valuable and very helpful for revising and improving our paper, as well as the important guiding significance to our researches. We have studied comments carefully and have made corrections which we hope meet with approval. In order to clearly show the changes of the manuscript to the editors and reviewers, the function of "Track Changes" in Microsoft Word was used, and the revised manuscript can be displayed properly by clicking the final status without markup in the review tab. The main corrections in the paper and the responds to the reviewer’s comments are as flowing:

Responds to the reviewer’s comments:

Open Review

Quality of English Language

( ) I am not qualified to assess the quality of English in this paper
( ) English very difficult to understand/incomprehensible
( ) Extensive editing of English language required
( ) Moderate editing of English language required
(x) Minor editing of English language required
( ) English language fine. No issues detected

Yes

Can be improved

Must be improved

Not applicable

Does the introduction provide sufficient background and include all relevant references?

( )

(x)

( )

( )

Are all the cited references relevant to the research?

(x)

( )

( )

( )

Is the research design appropriate?

(x)

( )

( )

( )

Are the methods adequately described?

(x)

( )

( )

( )

Are the results clearly presented?

(x)

( )

( )

( )

Are the conclusions supported by the results?

(x)

( )

( )

( )

Comments and Suggestions for Authors

The paper provides a good overview of the friction behavior of chromium-coated cemented carbide. In this study, a chromium coating was deposited on carbide substrates using a multiple arc ion plating process, and it was examined for adhesion strength, microhardness, and tribological properties.

The paper is well structured and very well written. There are minor spelling errors present, such as in line 86 "there were no obvious," and, Zr was swapped with Cr. The labels in the graphs should also be highlighted to make them more visible.

Response:

According to the Reviewer’s suggestion, the words "there were no obvious" in line 86 were changed to “there were not significantly”, and the word “Zr” in line 120 was changed to “Cr”.

The results of the paper are clear and well-developed. However, it would be beneficial to emphasize the ultimate application goal of the coated cemented carbides to make the results more applicable for future use.

Response:

According to the Reviewer’s suggestion, the related information about ultimate application goal of the Cr-coated cemented carbides was added in the end of result as follows:

Cr coating can be considered as effective way to enhance the friction and wear performance of traditional cemented carbide.

Comments on the Quality of English Language

There are minor spelling errors present, such as in line 86 "there were no obvious," and in line 120, Zr was swapped with Cr.

Response:

According to the Reviewer’s suggestion, the words "there were no obvious" in line 86 were changed to “there were not significantly”, and the word “Zr” in line 120 was changed to “Cr”.

We tried our best to improve the manuscript and made some changes in the manuscript.  These changes will not influence the content and framework of the paper. And here we did not list the changes but can be tracked down in revised paper by clicking "Track Changes".

We appreciate for your warm work earnestly, and hope that the correction will meet with approval.

Once again, thank you very much for your comments and suggestions.

Best Regards,

Corresponding author: Wenlong Song

Round 2

Reviewer 1 Report

A review must take into account the previous comments, I see there are not many improvements. Pre and posttreatments are well developed in manufacturing, in machining and you must check works published in that field.

Paper would be much better in the journal Coatings, because it is not about lubricants.

Major.

Author Response

Dear Reviewer:
Thank you for your letter and for the reviewer’ comments concerning our manuscript entitled “Fabrication and tribology property of Cr-coated cemented carbide under dry friction conditions” (ID: lubricants-2367373).

Comments and Suggestions for Authors

A review must take into account the previous comments, I see there are not many improvements. Pre and posttreatments are well developed in manufacturing, in machining and you must check works published in that field.

Response:

According to the Reviewer’s suggestion, the related information about pre and post treatments and some related literatures were added in the manuscript as follows:

To further improve the interfacial adhesive property between the substrate surface and coating, pre and post treatments technologies, such as sandblasting treatment, electrochemical corrosion, nitriding treatment, and laser treatment, are employed to activate and purify the material substrate surface. And among them, laser treatment has competitive advantages in coatings preparation, which can produce geometric textures with different morphologies, increase the surface contact area, good adhesion interface and mechanical locking ability for the coatings.

  1. Dong, B.Z.; Guo, X.H.; Zhang, K.D.; Zhang, Y.P.; Li, Z.H.; Wang, W.S.; Cai, C. Combined effect of laser texturing and carburizing on the bonding strength of DLC coatings deposited on medical titanium alloy. Surface & Coatings Technology 2022, 429, 127951.
  2. Hong, E.; Lee, H. Microstructure, bonding state and phase formation behavior of carbon-doped TiZrN coatings by laser carburization. Surface & Coatings Technology 2020, 385, 125373.
  3. Xing, Y.Q.; Wang, X.S.; Du, Z.YH.; Zhu, Z.W.; Wu, Z.; Liu L. Synergistic effect of surface textures and DLC coatings for enhancing friction and wear performances of Si3N4/TiC. Ceramics International 2022, 48, 514-524.
  4. Meng, Y.; Deng, J.X.; Lu, Y.; Wang, S.J.; Wu, J.X.; Sun, W. Fabrication of AlTiN coatings deposited on the ultrasonic rolling textured substrates for improving coatings adhesion strength. Applied Surface Science 2022, 550, 149394.

Paper would be much better in the journal Coatings, because it is not about lubricants.
Response:

Lubricants published papers covering all aspects of tribology, including the study and application of the principles of friction, lubrication and wear. Then, this manuscript is still within the scope of journal according to journal aim and scope.

We appreciate for your warm work earnestly. Once again, thank you very much for your comments and suggestions.

Best Regards,

Corresponding author: Wenlong Song

Round 3

Reviewer 1 Report

a